# Epidemiology of COVID-19 vs. influenza: Differential failure of COVID-19 mitigation among Hispanics, Cook County Health, Illinois

**William E. Trick**[1,2]*, **Sheila Badri**[3], **Kruti Doshi**[1], **Huiyuan Zhang**[1], **Katayoun Rezai**[3], **Michael J. Hoffman**[3], **Robert A. Weinstein**[2,3]

1 Center for Health Equity and Innovation, Health Research and Solutions, Cook County Health, Chicago, Illinois, United States of America, 2 Department of Medicine, Rush University Medical Center, Chicago, Illinois, United States of America, 3 Department of Medicine, Cook County Health, Chicago, Illinois, United States of America

* wtrick@cookcountyhhs.org

## Abstract

### Background

During the early phases of the COVID-19 pandemic in the U.S., African-American or Hispanic communities were disproportionately impacted. To better understand the epidemiology and relative effects of COVID-19 among hospitalized Hispanic patients, we compared individual and census-tract level characteristics of patients diagnosed with COVID-19 to those diagnosed with influenza, another viral infection with respiratory transmission. We evaluated temporal changes in epidemiology related to a shelter-in-place mandate.

### Methods

We evaluated patients hospitalized at Cook County Health, the safety-net health system for the Chicago metropolitan area. Among self-identified hospitalized Hispanic patients, we compared those with influenza (2019–2020 season) to COVID-19 infection during March 16, 2020-May 11, 2020. We used multivariable analysis to identify differences in individual and census-tract level characteristics between the two groups.

### Results

Relative to non-Hispanic blacks and whites, COVID-19 rapidly increased among Hispanics during promotion of social-distancing policies. Whereas non-Hispanic blacks were more likely to be hospitalized for influenza, Hispanic patients predominated among COVID-19 infections (40% relative increase compared to influenza). In the comparative analysis of influenza and COVID-19, Hispanic patients with COVID-19 were more likely to reside in census tracts with higher proportions of residents with the following characteristics: Hispanic; no high school diploma; non-US citizen; limited English speaking ability; employed in manufacturing or construction; and overcrowding. By multivariable analysis, Hispanic patients hospitalized with COVID-19 compared to those with influenza were more likely to be male (adjusted OR = 1.8; 95% CI 1.1 to 2.9), obese (aOR = 2.5; 95% CI 1.5 to 4.2), or

**Data Availability Statement:** All relevant data are available from https://dataverse.harvard.edu/api/access/datafile/4209394.

**Funding:** The authors received no specific funding for this work.

**Competing interests:** The authors have declared that no competing interests exist.

reside in a census tract with ≥40% of residents without a high-school diploma (aOR = 2.5; 95% CI 1.3 to 4.8).

## Conclusions

The rapid and disproportionate increase in COVID-19 hospitalizations among Hispanics after the shelter-in-place mandate indicates that public health strategies were inadequate in protecting this population—in particular, for those residing in neighborhoods with lower levels of educational attainment.

## Introduction

As of December 10, 2020 there have been over 1.5 million deaths due to COVID [1]. There have been many noteworthy contributions to the literature on the epidemiology of COVID-19 including comparisons of the burden of cases and mortality by race-ethnicity. During most of 2020, the U.S. emerged as the country with the highest absolute number of cases and deaths [2]. It is now well recognized that the dramatic increase in transmission after the first few months of the pandemic in the U.S. occurred disproportionately among non-Hispanic black and Hispanic communities [3]. Factors highlighted as potential drivers of this early and sustained transmission include overcrowding and limited ability to work remotely, leading to challenges in social distancing; multigenerational families; and a high-prevalence of comorbidities [4–6].

In the Cook County Health system, which provides care for the most vulnerable populations of the Chicago metropolitan region, predominantly non-Hispanic blacks and Hispanics, we noted an early surge in admissions of Hispanic patients. We sought to better understand the epidemiology and effect of COVID-19 in the Hispanic community by comparing patient- and census tract-level factors of patients hospitalized due to COVID-19 to those of patients hospitalized due to influenza infection, another viral infection with respiratory transmission, which historically has impacted non-Hispanic blacks with rates at least as high, if not higher, than Hispanics [7–9]. We evaluated temporal trends in COVID-19 by race-ethnicity and evaluated temporal changes in the context of social distancing policies and their resultant impact on mobility.

## Methods

The study was reviewed and approved by the institutional review board at Cook County Health. Informed consent was waived as this was an analysis of routinely collected data and was deemed to be of minimal risk. We assembled a cohort of patients hospitalized for influenza during the 2019–2020 influenza or for COVID-19 during the early phases of the pandemic, through May 11, 2020. We identified patients from a research data warehouse, which contains clinical data on all health system patients; ecological variables are captured through routine address cleaning and geocoding with linkage to U.S. census data [10]. We restricted the cohort to laboratory-confirmed influenza and COVID-19 cases; inclusion in the COVID-19 cohort was limited to hospitalized patients with SARS-CoV-2 detected by polymerase chain reaction —our first detected case was March 16, 2020. We focused on inpatients because persons hospitalized for respiratory infection during the influenza season routinely are evaluated for influenza; in contrast, laboratory confirmation of influenza in the emergency department is much less common.

For patient-level characteristics, we evaluated age, self-identified race-ethnicity; patient co-morbidities captured through ICD-10 diagnosis codes (we evaluated asthma, chronic kidney disease, congestive heart failure, COPD, diabetes, and HIV or AIDS,); and obesity, dichoto-mized as body mass index $\geq$30 kg/m$^2$. Ecological variables were included based on five-year estimates for census tracts published by the U.S. Census Bureau as the American Community Survey [10]. We made a priori selections of the following ecological variables to explore their association with COVID-19 infection: Overcrowding; employment in manufacturing, con-struction, or service occupations; not in labor force; preference for Spanish language; Hispanic ethnicity; high school diploma; poverty; native born in the U.S.; foreign born, non-citizen; internet access at home; computer at home; and a locally-calculated Social Vulnerability Index [11].

We evaluated the prevalence and associations between patient- and ecological-factors for COVID-19 versus influenza infection using bivariable analyses for all patient and ecological variables. We constructed multivariable logistic regression models to explore the association between COVID-19 and influenza. Because of multicollinearity between ecological variables, we separately entered each ecological variable into the final model to explore the strength of association in relatively parsimonious models. To more intuitively express the quantitative association between census tract variables and COVID-19 infection, we constructed a final model with dichotomous categorization of census tract variables into upper quartile vs the lower three quartiles based on the distribution of values in our dataset. To evaluate temporal trends across racial and ethnic groups, we segmented calendar years into weeks and calculated the proportion of hospitalization for Hispanics, non-Hispanic Blacks, and non-Hispanic whites. We constructed graphs over time using locally smoothed polynomial regression plots with 95% CI bands. All analyses were performed using Stata software, version 14.2.

## Results

The proportion of patients admitted to the hospital who were Hispanic was significantly higher for treatment of COVID-19 infection compared to influenza, (59% vs 42%; P<0.001), Fig 1.

The relative increase in COVID-19 infection among Hispanics became apparent by the third week after the initial COVID-19 hospitalization and continued to increase until reaching a plateau of over 50% of all patient admissions during Week 5, Fig 2. Among Hispanics, there

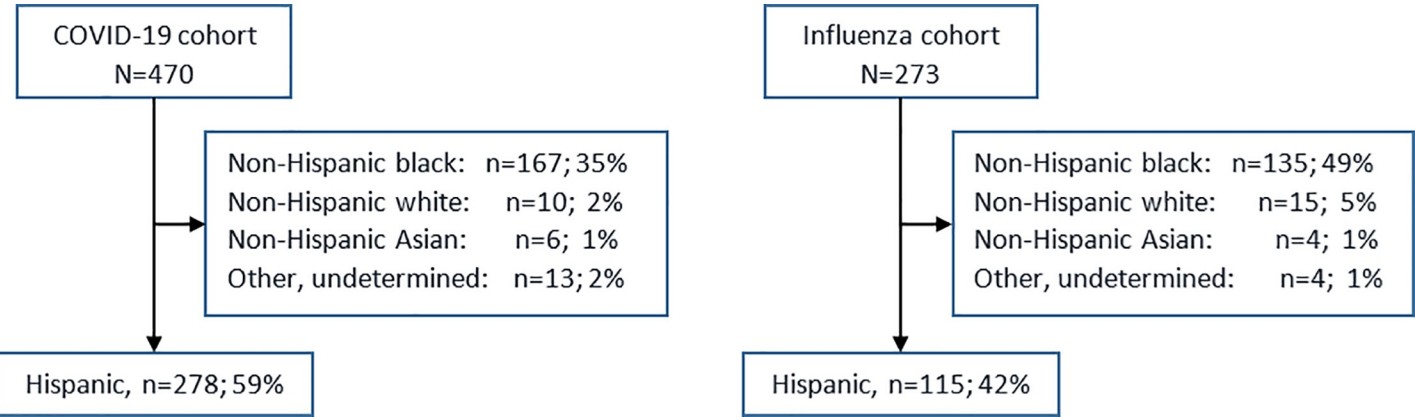

**Fig 1. Flow diagram depicting the COVID-19 and influenza infection cohorts stratified by race, Cook County Health, Chicago, Illinois, United States of America.**
[a] Hospital inpatients with a positive laboratory test for SARS-CoV-2 during March 16-May 11, 2020. [b] Hospital inpatients with a positive laboratory test for influenza virus during March 16-May 11, 2020.

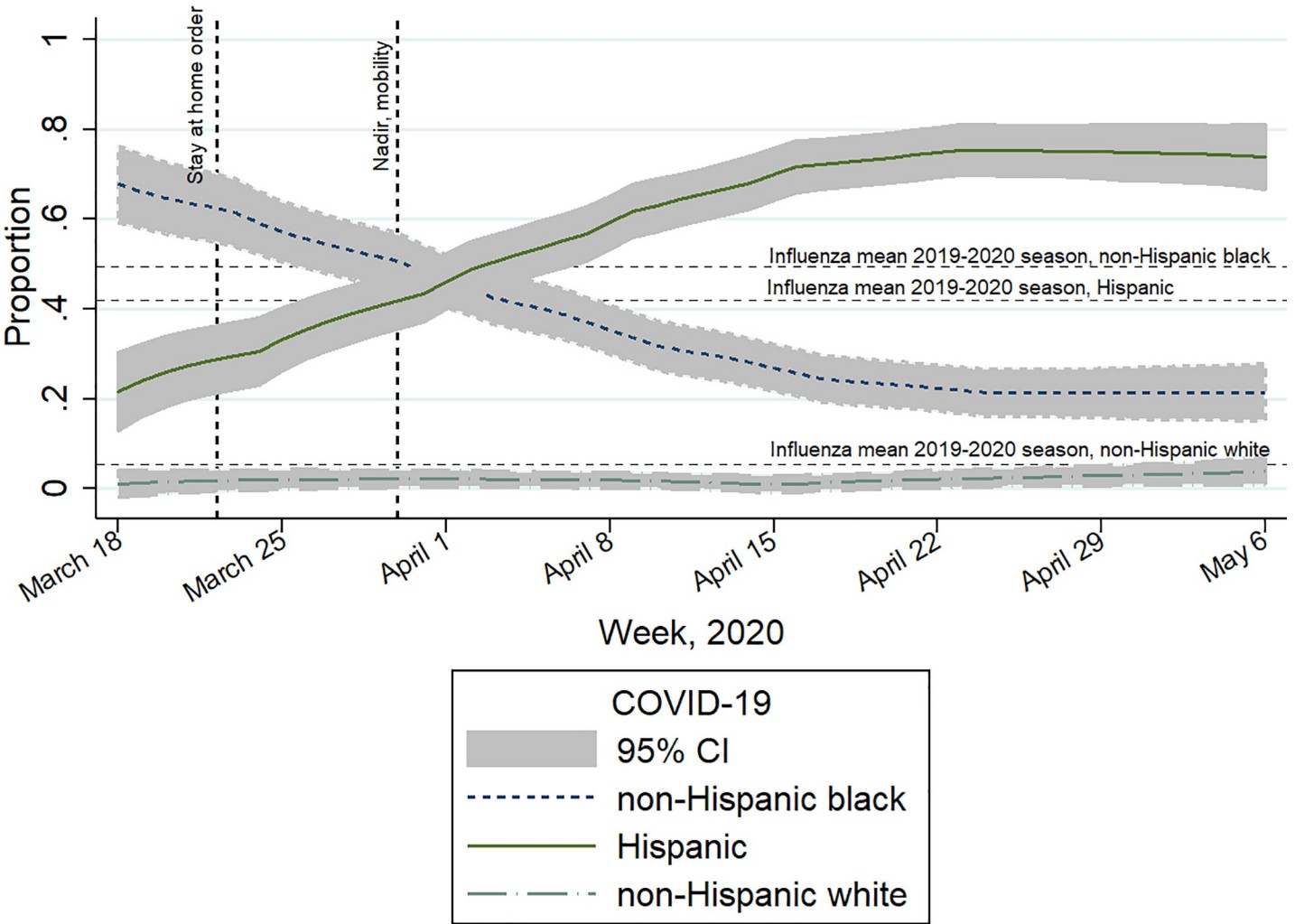

**Fig 2. Weekly trend in hospital admissions by race-ethnicity for COVID-19 compared to the mean race-ethnicity values for influenza during the 2019–2020 influenza season[a].** Findings presented in the context of social control policies and their resultant impact on mobility (represented by vertical lines), Cook County Health, Illinois. [a] There was a 42% relative increase in the proportion of hospitalized patients who were Hispanic compared to influenza infection.

were over two-fold more hospitalizations due to COVID-19 infection during our ~two-month study period (n = 278) than for influenza infection during the entire 2019–2020 influenza season (n = 115).

Among hospitalized Hispanic patients, when we evaluated patient-level factors associated with COVID-19 compared to influenza, COVID-19 patients were more likely to be male or obese, and less likely to have a diagnosis of asthma, chronic obstructive pulmonary disease (COPD), or heart failure, Table 1. Regarding patient outcomes, COVID-19 infected patients were more likely than those who acquired influenza to require admission to an ICU, and much more likely to require mechanical ventilation or die, Table 1.

Among hospitalized Hispanic patients, ecological factors significantly associated with COVID-19 compared to influenza infection were a higher proportion of the census tract population exhibiting the following: Hispanic ethnicity; Spanish language preference; not U.S. citizens; no high-school diploma; working in manufacturing or construction; and, overcrowding, Table 2.

**Table 1. Comparison of patient-level characteristics among hospitalized Hispanic patients with COVID-19 versus influenza infection[a], Cook County Health, Chicago, IL.**

| Covariates | COVID-19 (N = 278) | | Influenza (N = 115) | | Point Estimates | 95% CI | P-value |
|---|---|---|---|---|---|---|---|
| *Patient level Continuous* | Mean | SD | Mean | SD | **Difference** | | |
| Age, years mean (SD) | 52.9 | 12.7 | 54.6 | 15.6 | -1.7 | -4.7 to 1.3 | 0.26 |
| *Dichotomous* | n | % | n | % | **Prevalence Ratio** | | |
| Obese[b] | 132 | 55.7 | 35 | 34.3 | 1.6 | 1.2 to 2.2 | <0.001 |
| Male sex, n (%) | 182 | 65.5 | 63 | 54.8 | 1.2 | 1.0 to 1.4 | 0.05 |
| Diabetes mellitus | 141 | 50.7 | 59 | 51.3 | 1.0 | 0.8 to 1.2 | 0.92 |
| Asthma | 19 | 6.8 | 20 | 17.4 | 0.4 | 0.2 to 0.7 | 0.001 |
| Heart failure | 20 | 7.2 | 22 | 19.1 | 0.4 | 0.2 to 0.7 | <0.001 |
| COPD | 8 | 2.9 | 12 | 10.4 | 0.3 | 0.1 to 0.7 | 0.002 |
| *Outcomes* | | | | | | | |
| ICU admission | 78 | 28.1 | 21 | 18.3 | 1.5 | 1.0 to 2.4 | 0.04 |
| Ventilator use | 39 | 14.0 | 4 | 3.4 | 4.0 | 1.5 to 11 | 0.002 |
| In-hospital mortality | 37 | 13.3 | 0 | 0 | Undefined | | <0.001 |

Abbreviations: COPD, chronic obstructive pulmonary disease; SVI, social vulnerability index.

[a] Time periods: Influenza, 10/1/2019-5/11/2020; COVID-19, 3/16/2020-5/11/2020.

[b] Obese defined as Body Mass Index $\geq$30 kg/m$^2$.

By multivariable analysis, COVID-19 patients were more likely to be obese, male, or reside in a census tract where $\geq$40% of residents (upper quartile) reported no high-school diploma; and, less likely to have asthma or heart failure, Table 3.

Variable selection during multivariable model development was complicated by multi-collinearity among the ecological variables from census data, e.g., we found moderate to strong correlations between preference for Spanish-language and having no high-school diploma (correlation coefficient [rho] = 0.81), being a non-citizen (rho = 0.60), and employment in manufacturing (rho = 0.73) or construction (rho = 0.51).

## Discussion

By comparing rates of infection with COVID-19 to influenza, another epidemic respiratory disease, we confirmed previous reports that Hispanic patients experienced a disproportionate

**Table 2. Comparison of neighborhood characteristics of hospitalized Hispanic patients with COVID-19 versus influenza infection, Cook County Health, Chicago, IL.**

| Covariates[a] | COVID-19 (N = 278) | | Influenza[a] (N = 115) | | Point Estimates | 95% CI | P-value |
|---|---|---|---|---|---|---|---|
| | Mean % | SD | Mean % | SD | Difference | 95% CI | P-Value |
| Spanish preferred over English language | 64.1 | 26.7 | 57.0 | 27.5 | 7.1 | 1.2 to 13 | 0.02 |
| Hispanic | 71.3 | 28.6 | 64.4 | 27.3 | 7.0 | 0.9 to 13 | 0.02 |
| Non-citizen | 65.6 | 13.0 | 58.6 | 15.0 | 7.0 | 4.1 to 10 | <0.001 |
| No high school diploma | 33.0 | 13.7 | 28.2 | 12.1 | 4.8 | 1.9 to 7.6 | 0.001 |
| *Occupation* | | | | | | | |
| Manufacturing | 12.5 | 7.0 | 10.7 | 6.5 | 1.8 | 0.9 to 2.7 | <0.001 |
| Construction | 5.8 | 4.2 | 4.8 | 4.0 | 1.0 | 0.4 to 1.5 | <0.001 |
| Overcrowded | 8.5 | 5.6 | 7.3 | 4.4 | 1.2 | 0.4 to 2.3 | 0.04 |
| SVI | 77.4 | 16.6 | 74.6 | 17.3 | 2.8 | -0.8 to 6.8 | 0.13 |

Abbreviations: SVI, Social Vulnerability Index.

[a] We used five-year census-tract estimates from the US Census Bureau's American Community Survey.

**Table 3. Evaluation of ecological- and patient-level characteristics by multivariable analysis, COVID-19 versus influenza infection[a].** Hispanic patients hospitalized at Cook County Health, Chicago, IL.

| Variable | aOR | 95% CI | P-value |
|---|---|---|---|
| Obese | 2.5 | 1.5 to 4.2 | <0.001 |
| Census tract, ≥40% without HS diploma[b] | 2.5 | 1.3 to 4.8 | 0.007 |
| Male | 1.8 | 1.1 to 2.9 | 0.03 |
| Asthma | 0.4 | 0.2 to 0.9 | 0.02 |
| Heart failure | 0.3 | 0.2 to 0.7 | 0.001 |

Abbreviations: HS, High School

[a] Time periods: Influenza, 10/1/2019-5/11/2020; COVID-19, 3/16/2020-5/11/2020.

[b] Upper quartile for all census tracts represented by the study population.

burden of COVID-19 [12]. During the early phases of the pandemic, as social controls were promoted in the Chicago region through the local and state departments of public health, we observed a dramatic and sustained increase in the proportion of COVID-19 infections among Hispanics. The dramatic increase was temporally associated with Illinois' stay-at-home guidance and the closure of non-essential businesses enacted March 21, 2020. Associated with these policies was a resultant decrease in overall population mobility—a nadir of 56% reduced mobility was attained by March 29, 2020 [13].

The impact of social-distancing interventions would be expected to be effective primarily for individuals with capacity to "shelter-in-place", while transmission likely would continue unabated—possibly even accelerating—for individuals who were unable to "shelter-in-place" due to on-site employment or reduced job mobility due to legal status, occupation, and reduced opportunities associated with lower levels of educational attainment [14, 15]. Certain occupations have been associated with increased risk of COVID-19 infection and although we didn't collect individual patient occupation, our ecological analysis identified that among hospitalized Hispanic patients, COVID-19 was more likely than influenza infection to occur among residents of census tracts with a higher reported proportion of employment in construction and manufacturing. Additionally, the sociospatial characteristics, such as intense social ties, and multi-generational households that were described as protective for Hispanic communities during Chicago's 1995 heat wave, may have played a role in transmission of COVID-19 among social contacts in Hispanic communities outside of work [16].

Compared to influenza and not surprising, COVID-19 patients were more likely to have severe infection as manifested by a higher likelihood of ventilator use and mortality. The dramatic mortality difference is consistent with estimates of increased mortality risk from COVID-19 [17] and consistent with rates observed in other inpatient evaluations [18]. Although mortality could have been influenced by a considerable increase in the stress on the health system, the Chicago region, including Cook County Health, never experienced shortages of ICU beds or ventilators.

The increased risk of infection for males and for obesity has been described for both influenza and COVID-19 [19, 20]; however, these two characteristics were more common among COVID-19 compared to influenza patients. Our finding a lower prevalence of asthma or heart failure among hospitalized patients with COVID-19 infection should not be interpreted as an indicator that individuals with these chronic conditions are not at increased risk for COVID-19 compared to the general population. Rather, this finding may be explained by the higher virulence of COVID-19 compared to influenza; i.e., COVID-19 is more likely to result in illness severe enough to require hospitalization independent from pre-existing pulmonary or cardiac conditions.

In summary, in a health system that cares for patients who come from the most socially vulnerable communities, COVID-19 disproportionately affected Hispanic individuals and communities, a tragic situation that was foretold [21]. This suggests a differential and importantly disparate impact of mitigation measures across communities, especially among Hispanic patients. Additional studies are needed to better understand individual-level behaviors among Hispanics during the COVID-19 pandemic. For the current and future pandemics we need anticipatory plans that transcend community and individual risks, without which, disparities will result despite well-intentioned public health policies and interventions.

## Acknowledgments

We acknowledge Vanessa Sarda for her guidance on data analysis and presentation, and drafting of the manuscript.

## Author Contributions

**Conceptualization:** William E. Trick, Sheila Badri, Katayoun Rezai, Michael J. Hoffman, Robert A. Weinstein.

**Data curation:** Kruti Doshi, Huiyuan Zhang.

**Formal analysis:** William E. Trick.

**Methodology:** William E. Trick, Robert A. Weinstein.

**Software:** Kruti Doshi, Huiyuan Zhang.

**Writing – original draft:** William E. Trick.

**Writing – review & editing:** Sheila Badri, Kruti Doshi, Huiyuan Zhang, Katayoun Rezai, Michael J. Hoffman, Robert A. Weinstein.

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
