## [Decision Letter · Decision Letter 0]

30 Nov 2020

PONE-D-20-30228

Epidemiology of COVID-19 vs. Influenza: Differential Failure of COVID-19 Mitigation among Hispanics

PLOS ONE

Dear Dr. Trick,

Thank you for submitting your manuscript to PLOS ONE. After careful consideration, we feel that it has merit but does not fully meet PLOS ONE’s publication criteria as it currently stands. Therefore, we invite you to submit a revised version of the manuscript that addresses the points raised during the review process.

We look forward to receiving your revised manuscript.

Kind regards,

Muhammad Adrish

Academic Editor

PLOS ONE

Journal Requirements:

2. Please consider modifying your title to ensure that it is specific, descriptive, concise, and comprehensible to readers outside the field (for example by and including the name of the centre/location where you carried out the study)

3. In your Methods section, please provide additional information about the analysis performed, for example by reporting all the variables included in the analysis, and specifying how they were extracted and categorised.

4.We note that you have indicated that data from this study are available upon request. PLOS only allows data to be available upon request if there are legal or ethical restrictions on sharing data publicly. For information on unacceptable data access restrictions, please see http://journals.plos.org/plosone/s/data-availability#loc-unacceptable-data-access-restrictions.

Additional Editor Comments (if provided):

 I have received the comments of the reviewers on your manuscript. The specific comments of the reviewers are included below. Please provide point by point response in your revised manuscript.

Reviewers' comments:

Reviewer's Responses to Questions

**Comments to the Author**

1. Is the manuscript technically sound, and do the data support the conclusions?

Reviewer #1: Yes

Reviewer #2: Yes

2. Has the statistical analysis been performed appropriately and rigorously? 

Reviewer #1: Yes

Reviewer #2: Yes

3. Have the authors made all data underlying the findings in their manuscript fully available?

Reviewer #1: Yes

Reviewer #2: Yes

4. Is the manuscript presented in an intelligible fashion and written in standard English?

Reviewer #1: Yes

Reviewer #2: No

5. Review Comments to the Author

Reviewer #1: Dear Authors. I read with great interest your paper. I appreciate a lot the idea reaserch and the quality of manuscript. Only some minor suggestions:

1. Introduction: add dat on global burden of covid and deaths at the day of resubmission

2.Methods and results: well presented

3. Discussion: compare better your data with other data in literature (see and cite Common cardiovascular risk factors and in-hospital mortality in 3,894 patients with COVID-19: survival analysis and machine learning-based findings from the multicentre Italian CORIST Study. Nutr Metab Cardiovasc Dis. 2020 Oct 30;30(11):1899-1913. doi: 10.1016/j.numecd.2020.07.031.) and how covid impact not only for influenza but also for other fever diseases as malaria (see and cite Malaria and COVID-19: Common and Different Findings. Trop Med Infect Dis. 2020 Sep 6;5(3):141. doi: 10.3390/tropicalmed5030141.)

Reviewer #2: Trick et al, have written a clear and concise summary of what has been observed in Cook County. These data are well presented and would be interesting to compare to another county in a similar geolocation. The statistics capture what they summarise in their results. The second sentence of the results section is a little confusing, as it is unclear at first what the comparator groups are. I would recommend restructuring it to make clear whether you are looking at just hispanic populations with flu or the general number of cases of flu in line 2. In the methods section, you may want to capture what the total sample size was from which you derived the proportions of cases for the ethnicity sub study in a consort diagram or similar flow chart t show the work flow of your selection of patients to include, which led to Figure 1.

6. PLOS authors have the option to publish the peer review history of their article (what does this mean?). If published, this will include your full peer review and any attached files.

Reviewer #1: No

Reviewer #2: **Yes: **One B. Dintwe

---

## [Author Response · Author response to Decision Letter 0]

17 Dec 2020

Muhammad Adrish, Academic Editor, PLOS ONE December 8, 2020

Re: PONE-D-20-30228, Epidemiology of COVID-19 vs. Influenza: Differential Failure of COVID-19 Mitigation among Hispanics

Dear Dr. Adrish,

We appreciate the opportunity to revise and re-submit our manuscript “Epidemiology of COVID-19 vs. Influenza: Differential Failure of COVID-19 Mitigation for Hispanics” for consideration as a Research Article. Please see below for our replies, which are marked in bold font:

…please upload the minimal anonymized data set necessary to replicate your study findings as either Supporting Information files or to a stable, public repository and provide us with the relevant URLs, DOIs, or accession numbers. Please see http://www.bmj.com/content/340/bmj.c181.long for guidelines on how to de-identify and prepare clinical data for publication. For a list of acceptable repositories, please see http://journals.plos.org/plosone/s/data-availability#loc-recommended-repositories.

We have uploaded our dataset and a metadata file to Harvard Dataverse, the url is as follows: 

https://dataverse.harvard.edu/dataset.xhtml?persistentId=doi:10.7910/DVN/X1WNGL. 

We have binned the fields in our dataset to de-identify personal data in accord with privacy laws in consultation with our corporate compliance office. The url for the dataset is: 

Review Comments to the Author

Reviewer #1: Dear Authors. I read with great interest your paper. I appreciate a lot the idea reaserch and the quality of manuscript. Only some minor suggestions:

1. Introduction: add dat on global burden of covid and deaths at the day of resubmission

We have updated the numbers, which have changed dramatically.

2.Methods and results: well presented

3. Discussion: compare better your data with other data in literature (see and cite Common cardiovascular risk factors and in-hospital mortality in 3,894 patients with COVID-19: survival analysis and machine learning-based findings from the multicentre Italian CORIST Study. Nutr Metab Cardiovasc Dis. 2020 Oct 30;30(11):1899-1913. doi: 10.1016/j.numecd.2020.07.031.) and how covid impact not only for influenza but also for other fever diseases as malaria (see and cite Malaria and COVID-19: Common and Different Findings. Trop Med Infect Dis. 2020 Sep 6;5(3):141. doi: 10.3390/tropicalmed5030141.)

Thank you for pointing out this literature. I have included mortality estimates from the CORIST study, which was directly relevant to our manuscript. Although the paper on malaria was interesting to read, it was difficult to find the appropriate location in the manuscript to include this additional citation.

Reviewer #2: Trick et al, have written a clear and concise summary of what has been observed in Cook County. These data are well presented and would be interesting to compare to another county in a similar geolocation. The statistics capture what they summarise in their results. The second sentence of the results section is a little confusing, as it is unclear at first what the comparator groups are. I would recommend restructuring it to make clear whether you are looking at just hispanic populations with flu or the general number of cases of flu in line 2. 

We have modified the section to improve the clarity of the writing.

In the methods section, you may want to capture what the total sample size was from which you derived the proportions of cases for the ethnicity sub study in a consort diagram or similar flow chart t show the work flow of your selection of patients to include, which led to Figure 1.

Thank you for the suggestion, we have included a second figure, which is now Fig 1.

Sincerely,

William E. Trick, MD 

Director, Health Research & Solutions, Cook County Health

Co-Director, Center for Health Equity and Innovation 

Professor, Rush University Medical Center

1950 W. Polk St., Suite 5807

Chicago, IL 60612

Phone: 312-864-3631 Email: wtrick@cookcountyhhs.org

---

## [Decision Letter · Decision Letter 1]

2 Jan 2021

Epidemiology of COVID-19 vs. influenza: Differential failure of COVID-19 mitigation among Hispanics, Cook County Health, Illinois

PONE-D-20-30228R1

Dear Dr. Trick,

We’re pleased to inform you that your manuscript has been judged scientifically suitable for publication and will be formally accepted for publication once it meets all outstanding technical requirements.

Kind regards,

Muhammad Adrish

Academic Editor

PLOS ONE

Additional Editor Comments (optional):

Thank you for making the recommended revisions to your manuscript.

Reviewers' comments:

Reviewer's Responses to Questions

**Comments to the Author**

1. If the authors have adequately addressed your comments raised in a previous round of review and you feel that this manuscript is now acceptable for publication, you may indicate that here to bypass the “Comments to the Author” section, enter your conflict of interest statement in the “Confidential to Editor” section, and submit your "Accept" recommendation.

Reviewer #1: All comments have been addressed

Reviewer #2: All comments have been addressed

2. Is the manuscript technically sound, and do the data support the conclusions?

Reviewer #1: Yes

Reviewer #2: Yes

3. Has the statistical analysis been performed appropriately and rigorously? 

Reviewer #1: Yes

Reviewer #2: Yes

4. Have the authors made all data underlying the findings in their manuscript fully available?

Reviewer #1: Yes

Reviewer #2: Yes

5. Is the manuscript presented in an intelligible fashion and written in standard English?

Reviewer #1: Yes

Reviewer #2: Yes

6. Review Comments to the Author

Reviewer #1: authors improved they paper that now can be accepted. I appreciate a lot the manuscript and idea research that can improve the knowledge on COVID

Reviewer #2: (No Response)

7. PLOS authors have the option to publish the peer review history of their article (what does this mean?). If published, this will include your full peer review and any attached files.

Reviewer #1: No

Reviewer #2: **Yes: **One B. Dintwe

---

## [Editor Report · Acceptance letter]

19 Jan 2021

PONE-D-20-30228R1 

Epidemiology of COVID-19 vs. influenza: Differential failure of COVID-19 mitigation among Hispanics, Cook County Health, Illinois 

Dear Dr. Trick:

I'm pleased to inform you that your manuscript has been deemed suitable for publication in PLOS ONE. Congratulations! Your manuscript is now with our production department. 

Kind regards, 

on behalf of

Dr. Muhammad Adrish 

Academic Editor

PLOS ONE